# The allosteric activation of cGAS underpins its dynamic signaling landscape

**Richard M Hooy, Jungsan Sohn***

Department of Biophysics and Biophysical Chemistry, Johns Hopkins University School of Medicine, Baltimore, United States

**Abstract** Cyclic G/AMP synthase (cGAS) initiates type-1 interferon responses against cytosolic double-stranded (ds)DNA, which range from antiviral gene expression to apoptosis. The mechanism by which cGAS shapes this diverse signaling landscape remains poorly defined. We find that substrate-binding and dsDNA length-dependent binding are coupled to the intrinsic dimerization equilibrium of cGAS, with its N-terminal domain potentiating dimerization. Notably, increasing the dimeric fraction by raising cGAS and substrate concentrations diminishes duplex length-dependent activation, but does not negate the requirement for dsDNA. These results demonstrate that reaction context dictates the duplex length dependence, reconciling competing claims on the role of dsDNA length in cGAS activation. Overall, our study reveals how ligand-mediated allostery positions cGAS in standby, ready to tune its signaling pathway in a switch-like fashion.

DOI: https://doi.org/10.7554/eLife.39984.001

## Introduction

Whether arising endogenously or exogenously, double-stranded (ds)DNA in the cytoplasm of eukaryote cells indicates major problems (*Chen et al., 2016a*; *Paludan and Bowie, 2013*). For instance, genomic instability and damaged mitochondria introduce dsDNA into the cytoplasm (*Chen et al., 2016a*; *Denais et al., 2016*; *Mackenzie et al., 2017*; *Paludan and Bowie, 2013*; *Shen et al., 2015*; *West and Shadel, 2017*), and nearby rampant necrosis or pyroptosis can lead dsDNA to the cytoplasm of neighboring cells (*Abe et al., 2013*; *Ishii et al., 2001*). Moreover, the invasion of pathogenic bacteria or viruses introduces foreign dsDNA into the host cytoplasm (*Chen et al., 2016a*; *Paludan and Bowie, 2013*).

In metazoans, cyclic-G/AMP synthase (cGAS) plays a predominant role in initiating host innate immune responses against cytoplasmic dsDNA (*Chen et al., 2016a*; *Sun et al., 2013*). Upon detecting cytoplasmic dsDNA, cGAS cyclizes ATP and GTP into [2′−5′, 3′−5′]-linked cGAMP (*Gao et al., 2013b*), a unique host second-messenger for activating type-1 mediated stress-responses via Stimulator of Interferon Genes (STING). cGAS is integral not only to the host defense against all pathogens entailing DNA for replication (e.g. HIV, HSV, *L. monocytogenes*; (*Gao et al., 2013a*; *Hansen et al., 2014*; *Reinert et al., 2016*), but also to damaged organelles (*Mackenzie et al., 2017*; *West et al., 2015*). Moreover, cGAS plays a major role in regulating autoimmunity (e.g. Aicardi-Goutières syndrome and systemic lupus erythematosus [*An et al., 2017*; *Gao et al., 2015*; *Pokatayev et al., 2016*; *Vincent et al., 2017*]) and tumor formation and growth (*Ng et al., 2018*).

A signature of IFN-1 signaling is its variable outcomes, which include antiviral gene expression, cellular senescence, autophagy, and apoptosis (*Li and Chen, 2018*; *Li et al., 2016*; *Liang et al., 2014*; *Sun et al., 2013*; *van Boxel-Dezaire et al., 2006*; *Yang et al., 2017*). cGAS contributes significantly to this complex signaling landscape, with its signal strength, signaling duration, and cellular contexts influencing the type of outcomes (*Li and Chen, 2018*). For example, the outcome of the cGAS pathway depends on cell type (e.g. non-apoptotic macrophages vs. apoptotic T-cells

*For correspondence:
jsohn@jhmi.edu

Competing interests: The authors declare that no competing interests exist.

**eLife digest** The human immune system protects the body from various threats such as damaged cells or invading microbes. Many of these threats can move molecules of DNA, which are usually only found within a central compartment in the cell known as the nucleus, to the surrounding area, the cytoplasm.

An enzyme called cGAS searches for DNA in the cytoplasm of human cells. When DNA binds to cGAS it activates the enzyme to convert certain molecules (referred to as 'substrates') into another molecule (the 'signal') that triggers various immune responses to protect the body against the threat. To produce the signal, two cGAS enzymes need to work together as a single unit called a dimer.

The length of DNA molecules in the cytoplasm of cells can vary widely. It was initially thought that DNA molecules of any length binding to cGAS could activate the enzyme to a similar degree, but later studies demonstrated that this is not the case. However, it remains unclear how the length of the DNA could affect the activity of the enzyme, or why some of the earlier studies reported different findings.

Hooy and Sohn used biochemical approaches to study the human cGAS enzyme. The experiments show that cGAS can form dimers even when no DNA is present. However, when DNA bound to cGAS, the enzyme was more likely to form dimers. Longer DNA molecules were better at promoting cGAS dimers to form than shorter DNA molecules. The binding of substrates to cGAS also made it more likely that the enzyme would form dimers. These findings suggest that inside cells cGAS is primed to trigger a switch-like response when it detects DNA in the cytoplasm.

The work of Hooy and Sohn establishes a simple set of rules to predict how cGAS might respond in a given situation. Such information may aid in designing and tailoring efforts to regulate immune responses in human patients, and may provide insight into why the body responds to biological threats in different ways.

DOI: https://doi.org/10.7554/eLife.39984.002

[*Gulen et al., 2017*; *Larkin et al., 2017*; *Li et al., 2016*; *Tang et al., 2016*]), the amount of cGAMP (e.g. autophagy vs. apoptosis [*Gulen et al., 2017*; *Li et al., 2016*; *Liang et al., 2014*; *Tang et al., 2016*]), and the duration for which cells are stimulated with cGAMP (antiviral gene expression vs. apoptosis [*Gulen et al., 2017*; *Larkin et al., 2017*; *Li et al., 2016*; *Tang et al., 2016*]). The goal of the present study is to understand the molecular mechanisms by which cGAS drives such a dynamic signaling landscape.

Resting cGAS is thought to be an inactive monomer, and formation of a 2:2 dimer with dsDNA within the catalytic domain (human cGAS residue 157 – 522) is necessary for activation (2 cGAS molecules on two dsDNA strands [*Li et al., 2013*; *Zhang et al., 2014*]). cGAS recognizes dsDNA independent of sequence (*Gao et al., 2013b*; *Kranzusch et al., 2013*; *Li et al., 2013*; *Zhang et al., 2014*), thus it was initially proposed that any dsDNA long enough to support the dimerization of cGAS could activate the enzyme equally well (e.g. ~15 base-pairs, bps (*Chen et al., 2016a*; *Li et al., 2013*; *Zhang et al., 2014*)). However, it was long known that dsDNA of at least 45 bp was required to elicit IFN-1 responses in cells (*Chen et al., 2016a*; *Stetson and Medzhitov, 2006*; *Unterholzner et al., 2010*). Indeed, two recent studies demonstrated that cGAS discriminates against short dsDNA (*Andreeva et al., 2017*; *Luecke et al., 2017*). For instance, cGAS is minimally activated in cells by dsDNA shorter than 50 bps, and maximal activation requires dsDNA longer than 200 bps, with the length-dependence more pronounced at lower dsDNA concentrations (*Andreeva et al., 2017*; *Luecke et al., 2017*). The dependence on dsDNA length is thought to arise because cGAS dimers linearly propagate along the length of two parallel dsDNA strands without making inter-dimer contacts, consequently generating a ladder-like complex that increases the overall stability via avidity (*Andreeva et al., 2017*). Together, it is believed that dsDNA length-based signal-to-noise filtration occurs at the binding/recognition stage (i.e. different $K_D$s for different dsDNA lengths), but not at the signal transduction step (i.e. same $V_{max}$ for different dsDNA lengths (*Andreeva et al., 2017*)).

Our understanding of the mechanisms by which cGAS is activated has evolved over the years, yet it remains unclear why two conflicting views on the role of dsDNA length have existed. Moreover, we noted that neither the previous (dsDNA length-independent) nor current (dsDNA length-dependent) activation model provides a robust framework for understanding how cGAS might be able to shape its diverse signaling landscape. First, the relationship between dsDNA binding and activation is poorly established. For instance, it remains to be tested whether the initial dsDNA binding step alone sufficiently explains the dsDNA length-dependent activation of cGAS in cells. Second, the ladder model implies that dimerization efficiency continuously increases with dsDNA lengths (>1000 bps), while the optimal cellular response peaks with any dsDNA longer than ~200 bps (*Andreeva et al., 2017*). Third, the ladder model is heavily based on structural and functional studies of the catalytic domain of cGAS (cGAS$^{cat}$). It was recently proposed that the N-domain of cGAS binds dsDNA and plays a crucial role in its cellular function (*Tao et al., 2017*; *Wang et al., 2017*). Moreover, dsDNA binding by the N-domain is thought to enhance the activity of the monomeric enzyme, consequently lifting the dsDNA length restriction (*Lee et al., 2017*). Thus, it is not clear whether the ladder-like arrangement applies exclusively to cGAS$^{cat}$, or whether it is germane to the full-length protein (cGAS$^{FL}$). Finally, given that cGAS is the predominant sensor for cytoplasmic dsDNA (*Chen et al., 2016a*), it is imperative for this enzyme to amplify and attenuate its signaling cascade in a switch-like manner to ensure proper host responses. How cGAS achieves this important task remains poorly understood.

We find here that human cGAS can auto-dimerize without dsDNA. dsDNA regulates this intrinsic monomer-dimer equilibrium not only in a cooperative, but also in a length-dependent manner. Also unexpectedly, substrates (ATP/GTP) can pull cGAS into the dimeric state without dsDNA. Because ligand binding is coupled to dimerization, the length of dsDNA not only regulates binding and dimerization (signal recognition), but also the substrate binding and catalysis (signal transduction). Compared to cGAS$^{cat}$, cGAS$^{FL}$ auto-dimerizes more readily and also couples binding of both substrate and dsDNA to dimerization more efficiently, revealing a new function of the N-domain in potentiating the dimerization of cGAS. Dimerization is essential for dsDNA-mediated activation of both cGAS$^{FL}$ and cGAS$^{cat}$, and the dimers do not arrange in an ordered configuration on long dsDNA, suggesting the role of dsDNA length is to simply regulate the probability of dimerization. Importantly, shifting the monomer-dimer equilibrium via elevated enzyme and ATP/GTP concentrations in the absence of dsDNA does not override the requirement for dsDNA to activate cGAS. Instead, these other factors prime the enzyme to be activated even by short dsDNA, indicating that the dependence on duplex length can change according to cellular reaction context. Together, our results set forth a unifying activation model for cGAS in which the intrinsic monomer-dimer equilibrium poises the enzyme to dynamically turn on or off its signaling pathway in a switch-like fashion.

## Results

### Human cGAS$^{cat}$ can dimerize without dsDNA

Human cGAS$^{cat}$ (denoted as cGAS$^{cat}$ hereafter) eluted as two peaks in size-exclusion chromatography (SEC) depending on protein concentration (*Figure 1A*). With decreasing protein concentrations, the two peaks progressively merged into the one with the lower apparent molecular weight (*Figure 1A*), suggesting that cGAS$^{cat}$ is subject to an intrinsic monomer-dimer equilibrium without dsDNA (*Figure 1—figure supplement 1*). This was surprising, as previous studies showed that mouse cGAS$^{cat}$ behaved as a monomer (*Li et al., 2013*); we speculate that mouse-cGAS$^{cat}$ intrinsically dimerizes more weakly.

To further test the intrinsic dimerization capability of cGAS, we examined the oligomeric state using small-angle-x-ray-scattering (SAXS; *Figure 1B*). The radius of gyration (R$_g$) and the maximum diameter (D$_{max}$) for *apo*- cGAS$^{cat}$ at all tested concentrations aligned better with those of dsDNA-bound mouse-cGAS$^{cat}$ dimer (*Figure 1C–D*; [*Li et al., 2013*]). We analyzed the distrbution of monomeric and dimeric species using SAXS-estimated molecular weight (SAXS MoW2) and OLIGOMER in ATSAS (*Figure 1D* [*Mylonas and Svergun, 2007*; *Petoukhov et al., 2012*; *Petoukhov and Svergun, 2013*]). Here, the fraction of dimeric species was proportional to protein concentrations, and the dimerization constant was estimated to be ~20 µM (*Figure 1D*). Together, we concluded that cGAS has an intrinsic capacity to dimerize, albeit with low affinity.

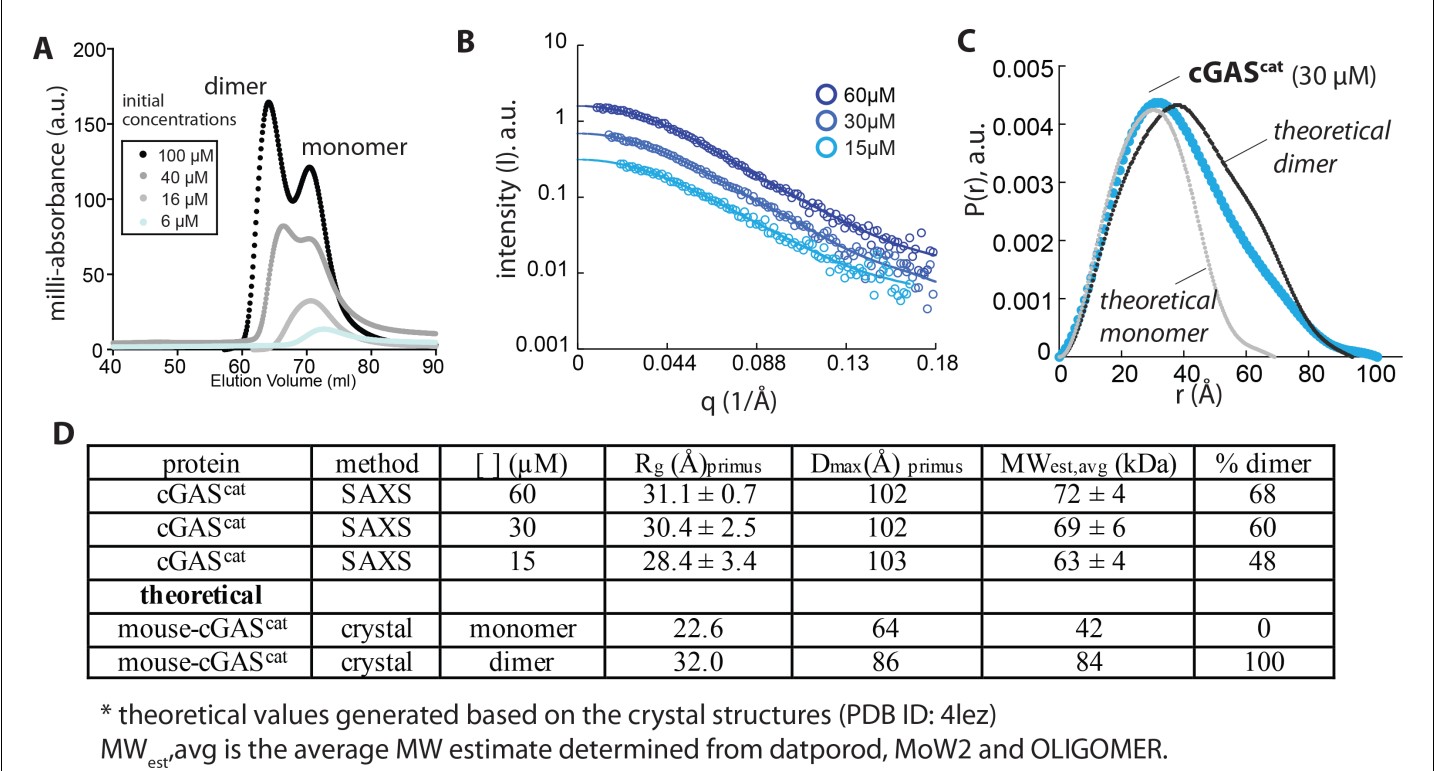

**Figure 1.** Human wild-type cGAS[cat] can dimerize on its own. (**A**) SEC (Superdex 75 16/600) profile of cGAS[cat]. (**B**) SAXS scattering profile of cGAS[cat]. (**C**) Pair-wise distance distribution functions of cGAS[cat]. Theoretical P(r)s from mouse-cGAS[cat] are shown for comparison (PDB ID: 4lez). (**D**) Summary of SAXS experiments.

DOI: https://doi.org/10.7554/eLife.39984.003

The following figure supplement is available for figure 1:

**Figure supplement 1.** Uv-vis absorbance profile of purified cGAS[cat].

DOI: https://doi.org/10.7554/eLife.39984.004

## cGAS behaves like a classic allosteric enzyme

In allosteric signaling enzymes, incoming signal (activator) and substrates either exclusively or preferentially bind to the active state and stabilize the corresponding conformation (*Koshland et al., 1966*; *Monod et al., 1965*; *Sohn et al., 2007*; *Sohn and Sauer, 2009*). Such a coupling mechanism synchronizes conformational states with activity states, thereby allowing the enzymes to generate switch-like responses (*Koshland et al., 1966*; *Monod et al., 1965*; *Sohn et al., 2007*; *Sohn and Sauer, 2009*). Importantly, preferential, but not exclusive ligand binding to the active state grades signaling output, as the distribution of active and inactive species is dictated by the relative binding affinity of different activators to either state (*Monod et al., 1965*; *Sohn and Sauer, 2009*; *Tsai and Nussinov, 2014*). Our observation that cGAS can dimerize on its own suggests a new framework for understanding its activation mechanism (*Figure 2A*). Here, *apo*-cGAS is placed in an intrinsic allosteric equilibrium where it is predominantly an inactive monomer under normal conditions. Overexpression (*Ma et al., 2015*), substrate binding, and cytoplasmic dsDNA synergistically activate cGAS by promoting dimerization. Furthermore, given that monomeric cGAS binds dsDNA (*Andreeva et al., 2017*; *Li et al., 2013*), it is possible that dsDNA length determines the fraction of active dimers (*Figure 2B*), thus underpinning the duplex length dependent cellular activity (*Andreeva et al., 2017*; *Luecke et al., 2017*). Below, we describe a series of experiments to further test and develop this allosteric framework for understanding the activation of cGAS.

The cellular activity of cGAS is dsDNA length-dependent (*Andreeva et al., 2017*; *Luecke et al., 2017*), as if the enzyme uses duplex length as a ruler to differentiate between signal and noise. Currently, it is believed that this length-based noise filtration occurs only at the initial encounter step,

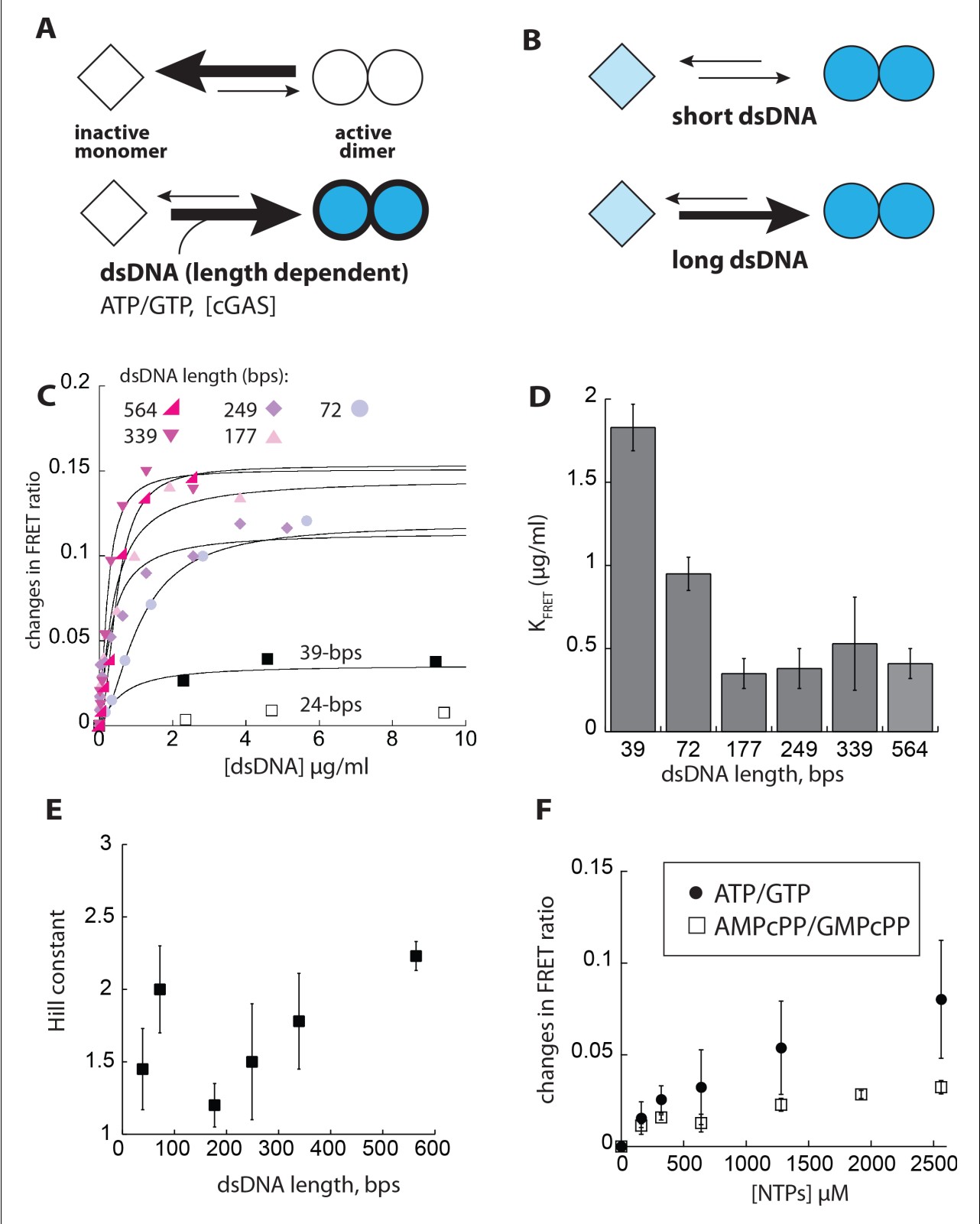

**Figure 2.** dsDNA cooperatively induces the dimerization of cGAS in a length-dependent manner. (**A**) A scheme describing the allosteric framework of cGAS activation. Here, cGAS is subject to an intrinsic allosteric equilibrium with two major activity/conformational states, namely inactive monomer and active dimer. Resting cGAS is predominantly an inactive monomer (top). dsDNA (length-dependent) binding, increasing cGAS concentration, and substrate binding synergistically drive the allosteric equilibrium toward the active dimer. (**B**) An allosteric model describing dsDNA length-dependent

*Figure 2 continued*

distribution of active dimers and inactive (basally active) monomers. (**C**) Changes in the ratio between FRET donor emission ($\lambda_{max}$: 578 nm) and the acceptor emission ($\lambda_{max}$: 678 nm) of labeled cGAS$^{cat}$ (20 nM each) at indicated dsDNA concentrations. (**D**) A plot of dimerization efficiency ($K_{FRET}$) vs. dsDNA length ($n = 3;\pm$SD). (**E**) A plot of fitted Hill constants vs. dsDNA lengths ($n = 3;\pm$SD). (**F**) Changes in the ratio between the FRET donor/acceptor emission ratios of labeled cGAS$^{cat}$ (20 nM each) at indicated NTP pair concentrations. ($n = 3;\pm$SD).

DOI: https://doi.org/10.7554/eLife.39984.005

The following figure supplement is available for figure 2:

**Figure supplement 1.** dsDNA binding and dimerization of cGAS.

DOI: https://doi.org/10.7554/eLife.39984.006

with longer dsDNA invoking a ladder-like arrangement (*Andreeva et al., 2017*). However, all previous binding studies entailed raising cGAS concentrations (*Andreeva et al., 2017*; *Li et al., 2013*), which intrinsically alters the dimer population. Thus, we re-examined the coupled relationship between dsDNA-binding and dimerization without altering the intrinsic dimerization equilibrium. First, using both direct and competition methods, we observed that cGAS$^{cat}$ indeed binds dsDNA in a length-dependent manner (*Figure 2—figure supplement 1A–B*). Next, to directly monitor dimerization, we conjugated a FRET donor and acceptor peptide to two populations of cGAS$^{cat}$ via sortaseA (FRET: Förster Resonance Energy Transfer; *Figure 2—figure supplement 1C*). The dimerization of a 1:1 mixture of donor- and acceptor-labeled cGAS$^{cat}$ at physiologically relevant concentrations was then tracked by changes in FRET emission ratios between the donor and acceptor with increasing concentrations of dsDNA (*Figure 2—figure supplement 1C*; physiological concentrations of cGAS vary between ~10 – 500 nM [*Andreeva et al., 2017*; *Du and Chen, 2018*; *Ma et al., 2015*]).

Increasing concentrations of 24 bp dsDNA did not induce significant changes in FRET ratios (*Figure 2C*), consistent with the previous report that such a short dsDNA binds cGAS but cannot induce dimerization (*Andreeva et al., 2017*). With longer dsDNA, we observed more robust changes in FRET signals (*Figure 2C*). Importantly, the half-maximal dsDNA concentrations necessary to induce the FRET signal ($K_{FRET}$) decreased with longer dsDNA, with the optimum length reaching at ~300 bps (*Figure 2C–D*). The maximal change in FRET ratio also generally increased with longer dsDNA, suggesting the dimeric fraction increased with longer dsDNA (*Figure 2C*). The fitted Hill constants in these experiments were between 1.5 and 2, indicating that dsDNA-induced dimerization is a cooperative process (*Figure 2E*). Overall, our results confirm that dsDNA binding and dimerization are directly coupled, consistent with the idea that the intrinsic monomer-dimer equilibrium underpins the dsDNA length discrimination by cGAS (*Figure 2A–B*).

It is thought that cGAS does not bind ATP/GTP in the absence of dsDNA, as the loops surrounding the active site would block substrate entry (*Gao et al., 2013b*). However, cGAS can bind cGAMP in the absence of dsDNA, and multiple crystal structures indicate that the B-factors of loops surrounding the active site are 5 to 20-fold higher than the protein core, suggesting cGAS might be able to weakly interact with ATP/GTP even without dsDNA (e.g. PDB IDs: 4k8v, 4o69, and 4km5; (*Gao et al., 2013b*; *Kranzusch et al., 2013*; *Zhang et al., 2014*)). Thus, we tested whether ATP/GTP and their nonhydrolyzable analogues (AMPcPP/GMPcPP) induce dimerization via our FRET assay. Here, introducing substrates increased the FRET ratio, albeit to a lower extent than long dsDNA (*Figure 2F*), suggesting that substrates alone can pull cGAS$^{cat}$ into the dimeric state to some degree. The lower capacity of AMPcPP/GMPcPP to induce FRET changes is consistent with our observations that the analogues bind more weakly than ATP/GTP ($K_i = 280$ µM (*Figure 2—figure supplement 1D*) vs. $K_M$ of ~100 for ATP/GTP with dsDNA, see *Figure 3* below). Together, our results suggest that the fraction of active, dimeric cGAS would be partitioned according to the length of dsDNA and the availability of substrates (*Figure 2A*). Thus, our results support that cGAS employs a strategy similar to classical allosteric enzymes to generate a graded output.

## A new quantitative assay for cGAS enzymatic activity

All published methods that quantitatively monitor the enzymatic activity of cGAS track cGAMP, and are not ideal for mechanistic studies due to their low throughput or difficulty in saturating the enzyme with substrates (e.g. TLC, HPLC-Mass-Spec, and fluorescently-labeled ATP/GTP; (*Andreeva et al., 2017*; *Gao et al., 2013b*; *Hall et al., 2017*; *Vincent et al., 2017*)). cGAS generates two inorganic pyrophosphates (PP$_i$) per cGAMP. Thus, we adapted a pyrophosphatase (PP$_i$ase)-

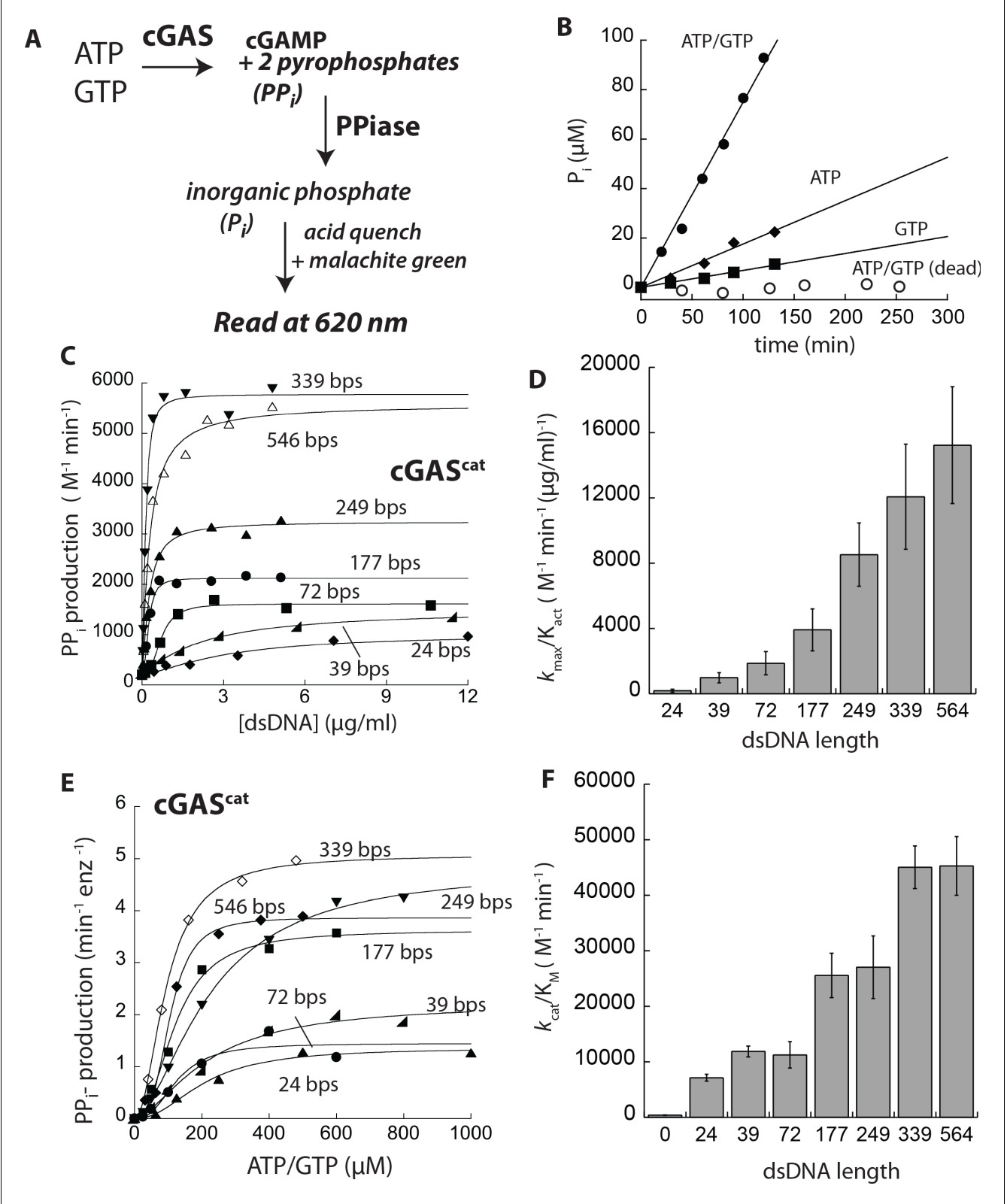

**Figure 3.** dsDNA length can grade the activity of cGAS. (**A**) PPiase-coupled assay scheme. (**B**) A plot of time-dependent $P_i$ production of cGAS$^{cat}$ (125 nM) at various conditions (dead: E225A-D227A). (**C**) A plot of the dsDNA concentration-dependent NTase activity of cGAS$^{cat}$ (25 nM) and 1 mM ATP/GTP with various duplex lengths. Lines are fits to a Hill form of the Michaelis-Menten equation. (**D**) A plot of dsDNA-affinity normalized maximal activities of cGAS$^{cat}$ vs. dsDNA lengths ($n$ = 3;±SD). (**E**) Substrate dependence of the steady-state rate of NTase activity by cGAS$^{cat}$ (125 nM) with

*Figure 3 continued on next page*

*Figure 3 continued*
saturating amounts of each dsDNA (6X $K_{act}$). Lines are fits to a Hill form of the Michaelis-Menten equation. (F) A plot of catalytic efficiencies ($k_{cat}/K_M$) vs. dsDNA lengths ($n$ = 3;±SD).
DOI: https://doi.org/10.7554/eLife.39984.007
The following figure supplements are available for figure 3:
**Figure supplement 1.** Kinetic parameters of cGAS and PP$_i$ase.
DOI: https://doi.org/10.7554/eLife.39984.008
**Figure supplement 2.** Summary of activation parameters.
DOI: https://doi.org/10.7554/eLife.39984.009

coupled assay developed by Stivers and colleagues (*Figure 3A*; (*Seamon and Stivers, 2015*)). Using this assay, we found that cGAS$^{cat}$ produces PP$_i$ most efficiently in the presence of a 1:1 mixture of ATP and GTP plus dsDNA (*Figure 3B*;>90% of its NTase activity produces cGAMP when ATP and GTP are equimolar (*Gao et al., 2013b*)). Moreover, no PP$_i$ production was observed from an inactive cGAS variant (E225A-D227A-cGAS$^{cat}$ (*Gao et al., 2013b*); *Figure 3B*), and the activity of PP$_i$ase was not rate-limiting (*Figure 3—figure supplement 1*). Thus, we concluded that the PP$_i$ase-coupled assay provides a robust method to quantitatively monitor the enzymatic activity of cGAS.

## dsDNA length regulates the extent of activation

Our experiments thus far support an activation model in which dsDNA length determines the distribution between active dimers and inactive monomers (*Figure 2A–B*). This mechanism entails different dsDNA lengths to produce graded maximal signaling output ($V_{max}$) even at saturating concentrations (*Sohn and Sauer, 2009*). In contrast, it has been proposed that the dsDNA length-dependent activity of cGAS arises solely at the signal recognition step (binding), but not at the signal transduction step (enzymatic step; (*Andreeva et al., 2017*)). However, the authors could not conduct their studies under steady-state conditions due to the use of fluorescently-labeled substrates (*Andreeva et al., 2017*). Because our coupled-assay eliminates this issue, we directly tested whether dsDNA length could regulate the enzymatic activity of cGAS. Here, we found that cGAS$^{cat}$ has low basal activity without dsDNA ($180 \pm 30$ M$^{-1}$min$^{-1}$), which can be increased by 50-fold with >300 bp dsDNA (*Figure 3C*). dsDNA concentrations required to induce the half-maximal activity of cGAS$^{cat}$ increased with shorter dsDNA ($K_{act}$; *Figure 3C* and *Figure 3—figure supplement 2A–B*), consistent with the previously observed length-dependent binding (*Andreeva et al., 2017*). Importantly, the maximum dsDNA-induced activity ($k_{max}$) also decreased with shorter dsDNA (*Figure 3C* and *Figure 3—figure supplement 2A and C*), which is in contrast to the previous report proposing that the role of dsDNA length is limited to binding (*Andreeva et al., 2017*). Moreover, normalizing the $k_{max}$ by $K_{act}$ for each dsDNA length showed that the overall signaling efficiency of cGAS$^{cat}$ (dsDNA binding and maximum output) changes more drastically compared to either parameter alone (*Figure 3D*, see also *Figure 3—figure supplement 2A–C*). For instance, the overall signaling efficiency changes by nearly 100-fold between 24 to 339 bp dsDNA, while either binding or maximal activity alone changes only up to 10-fold (*Figure 3D*, see also *Figure 3—figure supplement 2A–D*). Together, our observations indicate that cGAS discriminates against short dsDNA not only at the initial recognition step, but again at the signal transduction step, resulting in two-stage dsDNA length discrimination.

## dsDNA length regulates formation of the enzyme-substrate complex ($K_M$) and the turnover efficiency ($k_{cat}$) of cGAS

We next determined substrate turnover kinetics in the presence of various dsDNA lengths. Without dsDNA, cGAS$^{cat}$ showed measurable NTase activities (*Figure 3—figure supplement 1B*). With saturating dsDNA longer than 300 bps, the $K_M$ of cGAS$^{cat}$ for ATP/GTP was near 100 μM, and the $k_{cat}$ was 5 min$^{-1}$ (*Figure 3E* and *Figure 3—figure supplement 2D*). The observed $K_M$ for ATP/GTP is comparable to previously reported values measured using Surface Plasmon Resonance (SPR) and rapid-fire Mass-Spec for both human and mouse enzymes (*Hall et al., 2017*; *Vincent et al., 2017*). Moreover, the relatively slow $k_{cat}$ is consistent with a report indicating that human cGAS is considerably slower than mouse cGAS (~20 min$^{-1}$) (*Vincent et al., 2017*). Considering intracellular concentrations of ATP and GTP are >1 mM and ~500 μM, respectively (*Chen et al., 2016b*; *Traut, 1994*), our

result suggests that once cGAS encounters cytoplasmic dsDNA, one cGAMP would be generated in less than 20 s, compared to about one per 15 min in the absence of dsDNA. With shorter dsDNA, the $K_M$ increased about 2-fold, and the $k_{cat}$ decreased up to 4-fold (*Figure 3—figure supplement 2D*). Combined, our results indicated that the overall catalytic efficiency of cGAS can change up to 8-fold ($k_{cat}/K_M$) by the length of bound dsDNA (*Figure 3F* and Figure 3—figure supplement 3D). On another note, the fitted Hill constants in these experiments were near two for all dsDNA lengths (*Figure 3—figure supplement 2D*), consistent with the observation from mouse cGAS$^{cat}$ (*Vincent et al., 2017*). Because most cGAS$^{cat}$ populations would be dimeric with saturating long dsDNA, the observed cooperativity is likely from substrate-substrate interactions (i.e. ATP binding enhances GTP binding or *vice versa*; (*Vincent et al., 2017*)). Overall, these results further support that dsDNA length can grade the enzymatic activity of cGAS.

## The N-domain potentiates cGAS dimerization

It was recently reported that the N-domain of cGAS (residues 1 – 156) plays an important role in vivo by providing an additional nonspecific dsDNA binding site (*Tao et al., 2017*; *Wang et al., 2017*). Moreover, it was proposed that the N-domain reduces the requirement for long dsDNA, because it facilitates the activation of monomeric mouse cGAS (*Lee et al., 2017*). To test whether our findings using cGAS$^{cat}$ still apply to the full-length enzyme, we generated recombinant cGAS$^{FL}$. The full-length protein eluted as two peaks in SEC (*Figure 4A* and Figure), behaved as an extended particle by SAXS (*Figure 4B*, and *Figure 4—figure supplement 1B–C*), and was free from nucleic acid contamination (*Figure 4—figure supplement 1A*). Of note, it appeared that cGAS$^{FL}$ has a higher dimerization propensity compared to cGAS$^{cat}$, as indicated by broader peak distribution at 15 µM (*Figure 4A* vs. *Figure 1A*). Supporting this notion, SAXS analyses also suggested that the dimerization constant of cGAS$^{FL}$ is about 2-fold less than cGAS$^{cat}$ at ~7.5 µM (*Figure 4—figure supplement 1B–C*; cGAS$^{cat}$ is 48% dimeric at 15 µM; *Figure 1C–D*). To further test that the N-domain can dimerize we generated recombinant N-domain (cGAS$^N$) and found that it migrated as a dimer in SEC, and also behaved as an extended dimer in SAXS (*Figure 4—figure supplement 2A–C*). Of note, in our solution equilibrium assay, cGAS$^N$ bound dsDNA much more weakly than cGAS$^{cat}$, which is in contrast to the non-equilibrium mobility assay used by Tao et al. (*Tao et al., 2017*; *Figure 4—figure supplement 2D–E*; $K_D$ >10 µM). These observations suggest a new role of N-domain in assisting the dimerization of cGAS.

cGAS$^{FL}$ still bound dsDNA in a length dependent manner (*Figure 4—figure supplement 3A*), and displayed dsDNA length-dependent changes in FRET (*Figure 4C*, *Figure 4—figure supplement 3B*). $K_{FRET}$s for dsDNA > 72 bp were essentially the same under the minimal enzyme concentrations allowed in our assays (*Figure 4C*, *Figure 4—figure supplement 3B*), indicating that the full-length protein binds and dimerizes more readily on dsDNA. Substrates and their analogues also produced more robust changes in FRET signals for cGAS$^{FL}$ compared to cGAS$^{cat}$ (*Figure 4D*, *Figure 4—figure supplement 3B–C*), further corroborating that the full-length enzyme couples substrate binding to dimerization more efficiently due to its enhanced intrinsic dimerization activity. We also found that dsDNA length still grades $K_{act}$ and $k_{max}$ of cGAS$^{FL}$, as observed with cGAS$^{cat}$ (*Figure 4E*, *Figure 4—figure supplement 3D*); $K_M$ and $k_{cat}$ for cGAS$^{FL}$ were also graded according to dsDNA lengths (*Figure 4F*, *Figure 4—figure supplement 3E–F*). Overall, our observations indicate that cGAS$^{FL}$ and cGAS$^{cat}$ operate within the same molecular framework, and reveal a new role for the N-domain in potentiating the dimerization of cGAS.

## Dimerization is required for dsDNA-mediated activation

Although 24 bp dsDNA failed to induce dimerization (*Figures 2C* and *4C*), it activated cGAS to a significant extent (*Figures 3C* and *4E*). Monomeric cGAS can also bind dsDNA, but it is thought to be poorly activated (*Andreeva et al., 2017*; *Li et al., 2013*). Moreover, it was proposed that the N-domain enhances the dsDNA binding of monomeric cGAS (*Tao et al., 2017*), thereby activating the enzyme by lifting the dimerization requirement (*Lee et al., 2017*). Nonetheless, 24 bp dsDNA bound and activated both cGAS$^{cat}$ and cGAS$^{FL}$ only moderately (*Figures 3C* and *4E*). Thus, our data are most consistent with the allosteric model in which the presence of ATP/GTP increased the dimeric fraction, allowing the short dsDNA to activate cGAS to some extent (*Figure 2A–B*). To

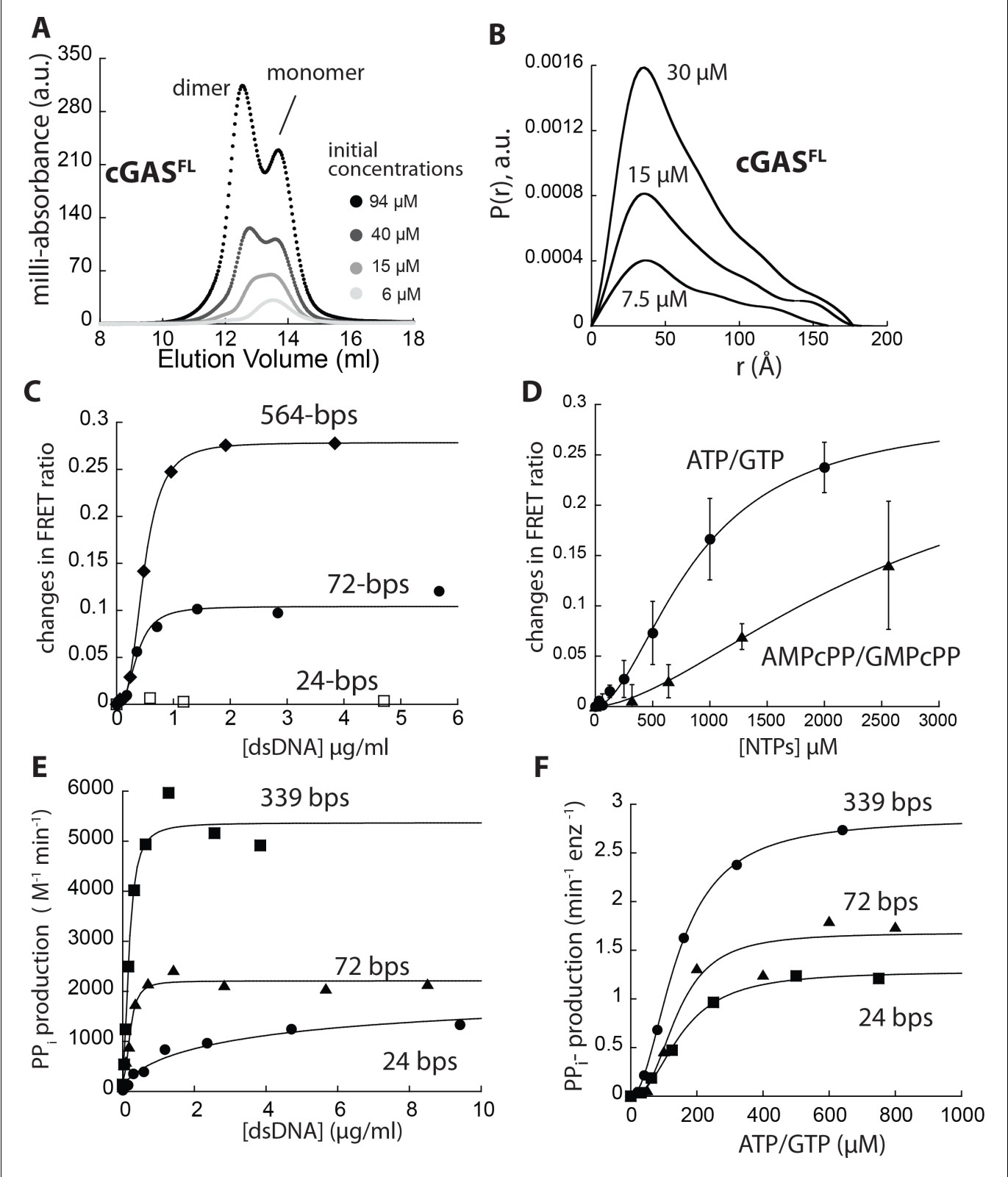

**Figure 4.** cGAS[FL] operates within the same allosteric framework as cGAS[cat]. (**A**) SEC of cGAS[FL] (Superdex 200 10/300). (**B**) Pair-wise distance distribution function of cGAS[FL]. (**C**) Changes in the ratio between the FRET donor emission ($\lambda_{max}$: 578 nm) and the acceptor emission ($\lambda_{max}$: 678 nm) of labeled cGAS[FL] (20 nM each) at indicated dsDNA concentrations. (**D**) Changes in the ratio between the FRET donor/acceptor emission ratio of labeled cGAS[FL] (20 nM each) at indicated NTP pair concentrations. (**E**) A plot of the dsDNA concentration-dependent NTase activity of cGAS[FL] (25 nM) and 800

*Figure 4 continued on next page*

*Figure 4 continued*

µM ATP/GTP with various duplex lengths. Lines are fits to a Hill form of the Michaelis-Menten equation. (F) Substrate dependence of the steady-state rate of NTase activity by cGAS$^{FL}$ (125 nM) with saturating amounts of each dsDNA (6X $K_{act}$). Lines are fits to a Hill form of the Michaelis-Menten equation.

DOI: https://doi.org/10.7554/eLife.39984.010

The following figure supplements are available for figure 4:

**Figure supplement 1.** Biophysical properties of cGAS$^{FL}$.

DOI: https://doi.org/10.7554/eLife.39984.011

**Figure supplement 2.** Biophysical properties of cGAS$^{N}$.

DOI: https://doi.org/10.7554/eLife.39984.012

**Figure supplement 3.** dsDNA binding and dimerization activities of cGAS$^{FL}$.

DOI: https://doi.org/10.7554/eLife.39984.013

further test this idea, we characterized the activities of a cGAS mutant that binds dsDNA but fails to dimerize, K394E (*Li et al., 2013*; *Zhang et al., 2014*; *Figure 5A*.

Both K394E-cGAS$^{cat}$ and K394E-cGAS$^{FL}$ behaved as single monomeric species in SEC (*Figure 5B* and *Figure 5—figure supplement 1A*), consistent with previous reports (*Li et al., 2013*; *Zhang et al., 2014*). SAXS experiments corroborated that K394E-cGAS$^{cat}$ is predominantly monomeric at all tested concentrations (*Figure 5—figure supplement 1B–D*). Compared to wild-type, not only did K394E-cGAS$^{cat}$ bind dsDNA more weakly, but also without length dependence (*Figure 5—figure supplement 2A*). The dsDNA length-dependence of K394E-cGAS$^{FL}$ was also less pronounced compared to wild-type cGAS$^{FL}$ (*Figure 5—figure supplement 2B–C*). We predict that the dsDNA length dependence of K394E-cGAS$^{FL}$ likely arise from the dimerization of the N-domain. Importantly, without dsDNA, K394E-cGAS showed similar activities as wild-type; however, dsDNA failed to stimulate the enzymatic activity of the mutants regardless of duplex length (*Figure 5C–F*). For instance, dsDNA marginally decreased the $K_M$ of K394E-cGAS, but the $k_{cat}$ did not increase significantly (*Figure 5D and F*). Our results also support the idea that monomeric cGAS can bind substrate and is basally active, yet dimerization is necessary for dsDNA- and dsDNA length-dependent activation regardless of the intact N-domain. Furthermore, our observations support the idea that short dsDNA and substrates can synergistically activate cGAS (see also Figure 7 below).

## cGAS dimers appear to arrange randomly on dsDNA

cGAS dimers are thought to form a ladder-like array along the length of dsDNA to maximize the stability of its signaling complex (*Andreeva et al., 2017*). Given that both cGAS monomers and dimers bind dsDNA (*Andreeva et al., 2017*; *Li et al., 2013*), our results are better explained by a simpler mechanism in which dsDNA length regulates the fraction of cGAS dimers without invoking an ordered structure (*Figure 2A–B*). To further test this idea, we imaged cGAS$^{cat}$ and cGAS$^{FL}$ with dsDNA using nsEM (*Figure 6*; see also *Figure 6—figure supplement 1* for zoom-in images, and additional images in *Figure 6—figure supplement 2*). When proteins were in excess over dsDNA, we observed large clusters likely reflecting multiple cGAS dimers binding to several different dsDNA strands (*Figure 6A and E*). It is possible that these clusters reflect the recently observed phase-shifting condensates of cGAS•dsDNA (*Du and Chen, 2018*). With excess dsDNA over protein, which more likely resembles in vivo events when dsDNA breaches the cytoplasm, it appeared that cGAS dimers randomly decorated dsDNA (*Figure 6B and F*), with the particle sizes corresponding to the dimeric species of cGAS$^{cat}$ and cGAS$^{FL}$, respectively (i.e. the $D_{max}$ for these constructs are ~10 and 18 nm, respectively; *Figure 1*). Importantly, the ladder-like arrangement of cGAS particles was rare for both cGAS$^{cat}$ and cGAS$^{FL}$ (*Figure 6B and F*, *Figure 6—figure supplement 2D–E*), suggesting that cGAS•dsDNA does not form an ordered supra-structure.

On the other hand, the size of particles resulting from excess K394E-cGAS$^{cat}$ with dsDNA appeared smaller and corresponded to the $D_{max}$ of cGAS monomers (*Figure 6C*; see also *Figure 5—figure supplement 1*), likely reflecting monomeric cGAS randomly bound on dsDNA. For K394E-cGAS$^{FL}$, we observed dsDNA-bound clusters somewhat similar to wild-type (these clusters are likely mediated by the intact N-domain that promotes dimerization). However, the clusters were not as expansive as those formed by wild-type (*Figure 6E* vs. G). Moreover, we did not observe any significant decoration of dsDNA when the K394E mutants were present in sub-stoichiometric amounts

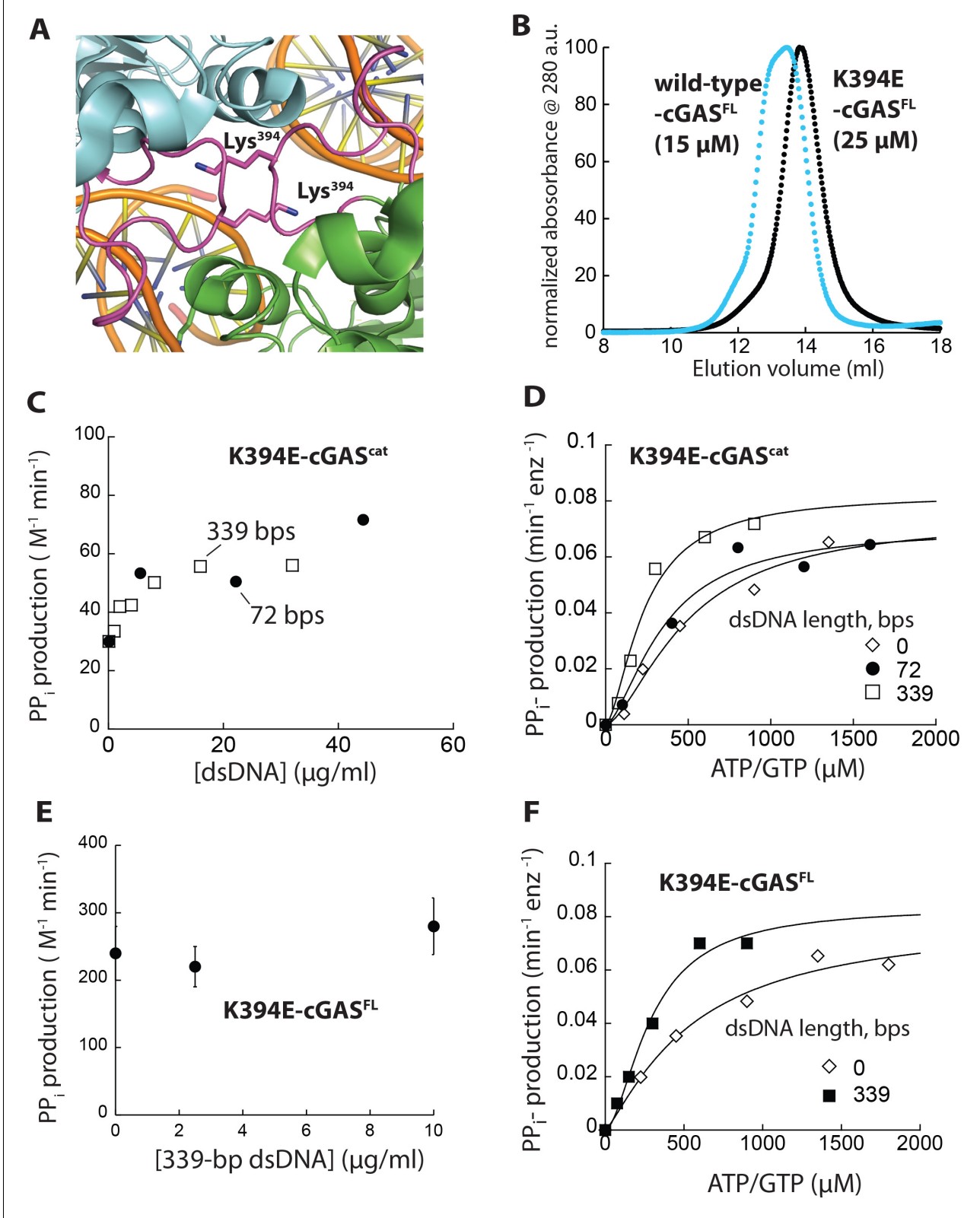

**Figure 5.** Monomeric cGAS is basally active, but cannot be further activated by dsDNA. (**A**) Crystal structure of dimeric cGAS^cat. The loop important for dimerization is colored in magenta and Lys^394 is shown in stick representation (PDB ID: 4lez). (**B**) SEC (Superdex 200 10/300) of K394E-cGAS^FL. WT-cGAS^FL is shown for reference (blue). (**C**) A plot of the dsDNA-concentration dependent NTase activity of K394E-hcGAS^cat (1 μM) and 1 mM ATP/GTP with different duplex lengths. (**D**) Substrate dependence of the steady-state rate of NTase activity by K394E-hcGAS^cat (1 μM) in the absence or presence

*Figure 5 continued on next page*

*Figure 5 continued*

of each dsDNA (6X K$_{act}$). Lines are fits to a Hill form of Michaelis-Menten equation. (**E**) A plot of the dsDNA-concentration dependent NTase activity of K394E-cGAS$^{FL}$ (125 nM) and 1 mM ATP/GTP with different duplex lengths. (**F**) Substrate dependence of the steady-state rate of NTase activity by K394E-cGAS$^{FL}$ (125 nM) in the absence or presence of each dsDNA (6X K$_{act}$). Lines are fits to a Hill form of Michaelis-Menten equation.

DOI: https://doi.org/10.7554/eLife.39984.014

The following figure supplements are available for figure 5:

**Figure supplement 1.** Biophysical properties of K394E-cGAS.

DOI: https://doi.org/10.7554/eLife.39984.015

**Figure supplement 2.** dsDNA binding properties of K394-cGAS.

DOI: https://doi.org/10.7554/eLife.39984.016

(*Figure 6D and H*; the particle size observed in *Figure 6H* also corresponds to the monomeric full-length cGAS). Overall our nsEM experiments support the allosteric framework of cGAS (*Figure 2A–B*) in which the role of dsDNA length is to simply bias the fraction of active dimers without necessitating supramolecular assemblies. Nevertheless, given the low-resolution imaging of nsEM, future structural studies are warranted to more fully understand the nature of these cGAS•DNA complexes.

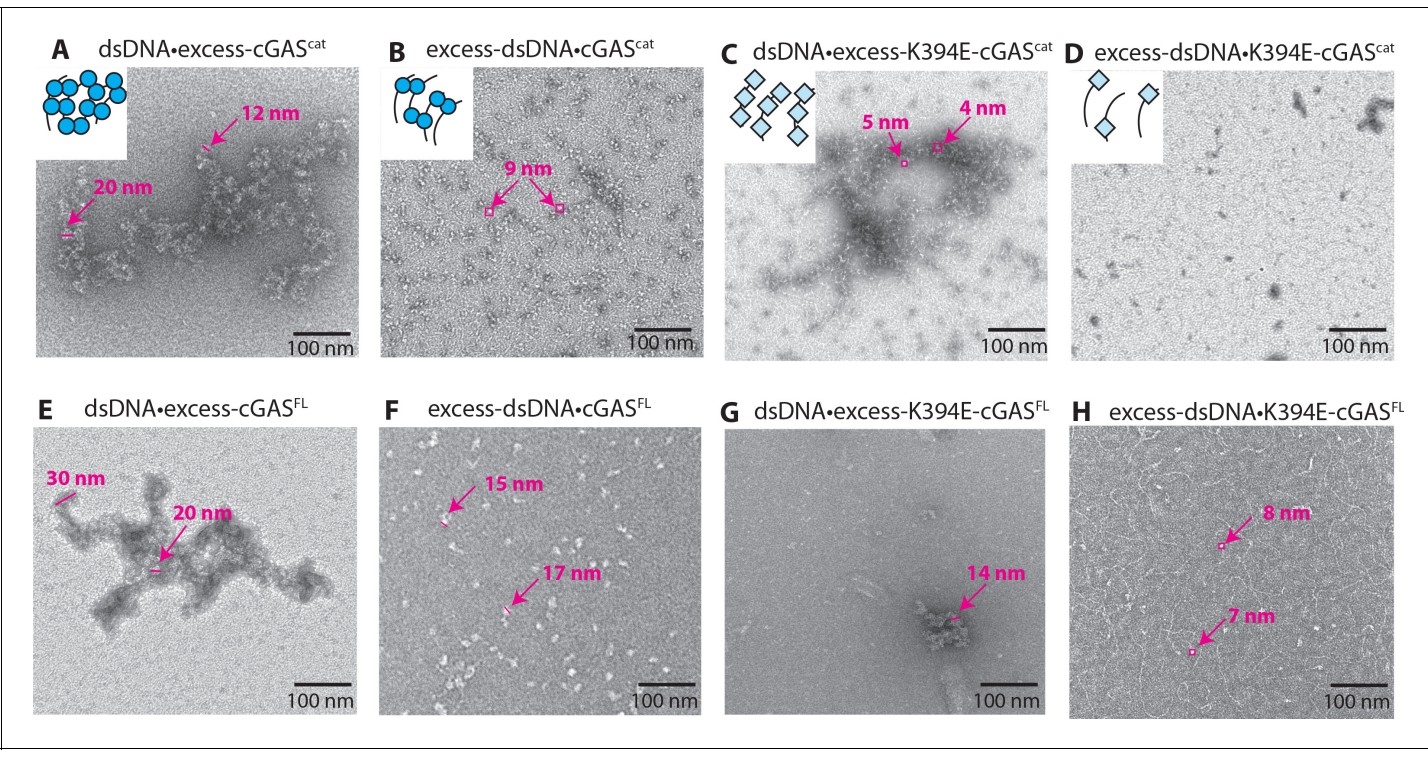

**Figure 6.** cGAS dimers assume various configurations on dsDNA. Negative-stain electron micrographs of cGAS•dsDNA complexes. (**A, E**) 3-fold excess cGAS over dsDNA, (**B, F**) 3-fold excess dsDNA over cGAS, (**C, G**) 3-fold excess K349E-cGAS over dsDNA, (**D, H**) 3-fold excess dsDNA over K394E-cGAS. Ratios of protein to dsDNA or dsDNA to protein are binding site normalized; 18 bp per binding site. The particle sizes in B and F are consistent with the D$_{max}$ of cGAS$^{cat}$ and cGAS$^{FL}$, respectively (*Figures 1* and *4*). Particle sizes in C and H are consistent with the D$_{max}$ of K394E-cGAS variants (*Figure 5—figure supplement 1*).

DOI: https://doi.org/10.7554/eLife.39984.017

The following figure supplements are available for figure 6:

**Figure supplement 1.** Additional nsEM images.

DOI: https://doi.org/10.7554/eLife.39984.018

**Figure supplement 2.** Additional nsEM images.

DOI: https://doi.org/10.7554/eLife.39984.019

## The context-dependent, allosteric activation of cGAS

It was initially proposed that dsDNA length does not play a significant role in regulating the activation of cGAS (*Gao et al., 2013b*; *Kranzusch et al., 2013*; *Li et al., 2013*); however, two recent studies have contested this model (*Andreeva et al., 2017*; *Luecke et al., 2017*). The reason for this discrepancy is still unclear. Our results suggest that raising enzyme and substrate concentrations increases the dimeric fraction of cGAS, while binding of short dsDNA cannot (e.g. 24 bp). Given the vastly different cGAS and substrate concentrations used in previous studies (*Andreeva et al., 2017*; *Gao et al., 2013b*; *Kranzusch et al., 2013*; *Li et al., 2013*; *Luecke et al., 2017*), we speculated that the apparent or lack of dsDNA length-dependence is caused by the fraction of cGAS dimers formed without dsDNA (*Figure 2A*). To test this idea, we monitored the steady-state NTase activity of cGAS$^{cat}$ and cGAS$^{FL}$ with saturating amounts of various dsDNA lengths and a permutation of high and low concentrations of enzyme and ATP/GTP (*Figure 7A–D*). Increasing substrate and enzyme concentrations did not eliminate the need for dsDNA. However, the dependence on dsDNA length progressively decreased with increasing protein and substrate concentrations. For instance, with low cGAS$^{cat}$ and sub-$K_M$ ATP/GTP concentrations (cGAS is predominantly monomeric), we observed strong dsDNA length-dependent activities, with a difference of 8-fold between 24 bp and 564 bp dsDNA (*Figure 7A*). With low cGAS and high ATP/GTP, the difference between short and long dsDNA was 4-fold (*Figure 7B*). With high cGAS and low ATP/GTP, the difference was again reduced to 2.5-fold (*Figure 7C*). Finally, with high cGAS$^{cat}$ and high ATP/GTP (the dimer population is significant), the differential activity caused by various dsDNA lengths was merely 1.5-fold, with short dsDNA molecules robustly activating cGAS$^{cat}$ (*Figure 7D*). Furthermore, we observed the same trend from cGAS$^{FL}$ except the effect of raising substrate and enzyme concentrations was more pronounced than cGAS$^{cat}$ (*Figure 7—figure supplement 1*). These observations uncover the reason for conflicting observations regarding dsDNA length-dependence (*Andreeva et al., 2017*; *Kranzusch et al., 2013*; *Li et al., 2013*; *Luecke et al., 2017*). That is, the dependence on dsDNA length can either manifest or diminish by different reaction contexts that dictate the fraction of dsDNA-free cGAS dimers. Our results in turn indicate that cGAS is primed to generate a graded signaling output depending on the overall reaction condition (e.g. the length of cytoplasmic dsDNA, cGAS expression level, and available ATP/GTP), providing a molecular framework for its context-dependent and diverse stress responses (*Gulen et al., 2017*; *Larkin et al., 2017*; *Li and Chen, 2018*; *Li et al., 2016*; *Tang et al., 2016*)

## Discussion

The activation of IFN-1 leads to diverse stress responses (antiviral gene expression, cellular senescence, autophagy, or apoptosis; (*Gulen et al., 2017*; *Larkin et al., 2017*; *Li and Chen, 2018*; *Li et al., 2016*; *Liang et al., 2014*; *Tang et al., 2016*; *Yang et al., 2017*)). cGAS contributes significantly to this complex signaling landscape by generating variable amounts of cGAMP (*Li and Chen, 2018*). Here, building upon the framework shown in *Figure 2A*, we set forth a unifying allosteric activation mechanism of cGAS, which explains how this cytoplasmic dsDNA sensor could dynamically tune its signaling activity in a switch-like fashion according to reaction (cellular) contexts (*Figure 7E*). In this model, cGAS is subject to an intrinsic monomer-dimer equilibrium, with its N-domain potentiating the dimerization propensity. dsDNA can drive the monomer-dimer equilibrium toward the dimeric state, with duplex length determining the fraction of active dimers (*Figure 7E* upper right-hand path). Importantly, given the active unit of cGAS is a dimer, we propose that longer dsDNA simply increases the probability of forming dimers without invoking an ordered configuration. We also find here that cGAS allosterically couples its dimeric population to factors other than dsDNA, such as cGAS expression level and ATP/GTP availability (*Figure 7E*, left path). We propose that this coupling mechanism would allow the dimer population to be in constant flux, providing a molecular framework for its dynamic signaling activity. Indeed, cGAS is subject to overexpression by multiple factors including its downstream product IFN-1 (*Ma et al., 2015*). Intracellular ATP/GTP concentrations also vary depending on cell age, cell-cycle progression, and stress conditions (*Corton et al., 1994*; *Huang et al., 2003*; *Marcussen and Larsen, 1996*; *Traut, 1994*; *Wang et al., 2003*). Moreover, post-translational modification (e.g. mono-ubiquitination) promotes dimerization of cGAS (*Seo et al., 2018*). Of note, given that pathogen infection increases host NTP levels (*Chang et al., 2009*; *Ogawa et al., 2015*), it is tempting to speculate that cGAS takes advantage of the higher

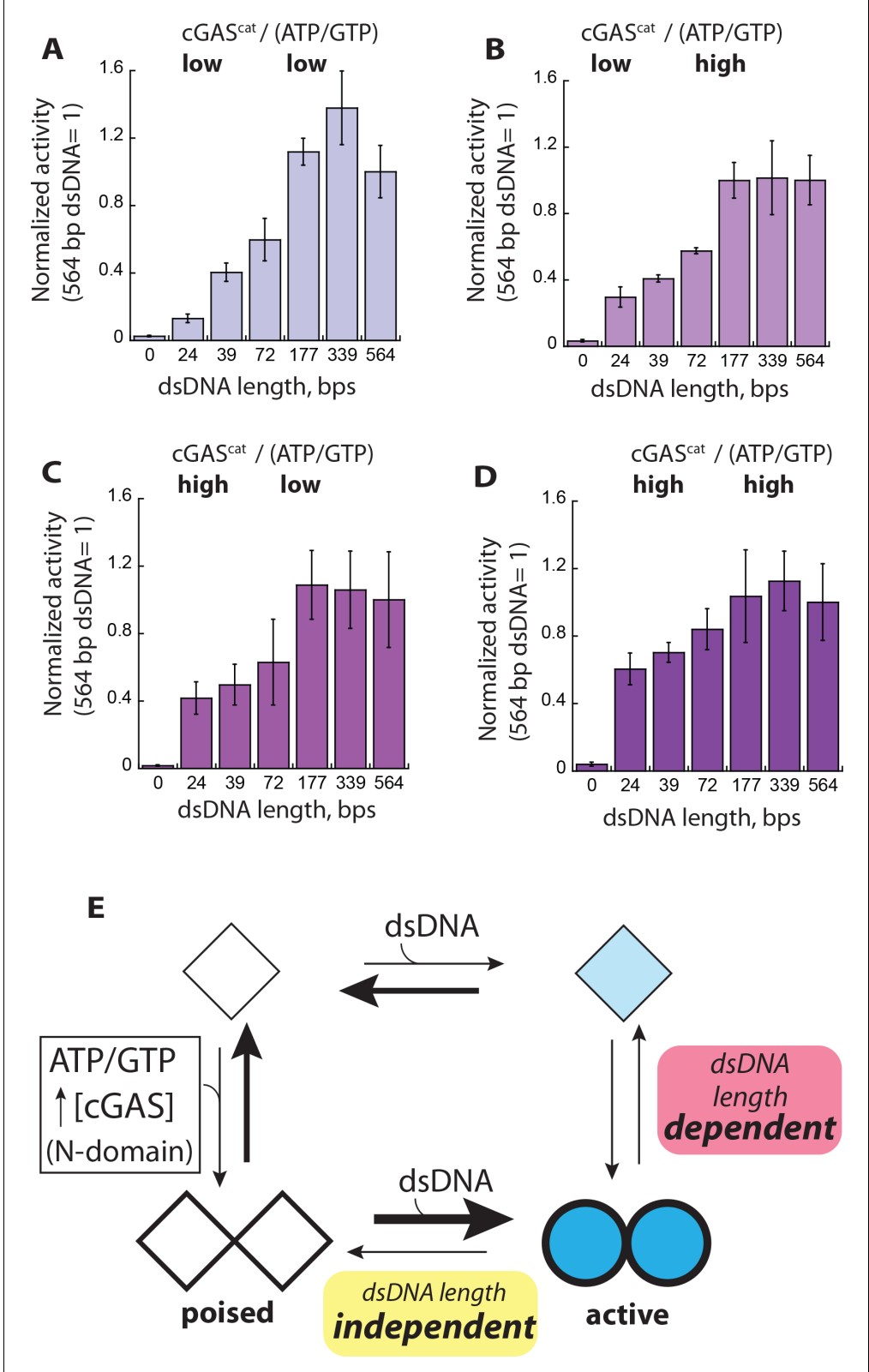

**Figure 7.** Context dependent activation of cGAS. (**A–D**) Plots of the NTase activity of cGAS$^{cat}$ vs. dsDNA length (6X-K$_{act}$) at various enzyme and substrate concentrations (low cGAS: 50 nM, High cGAS: 1 μM; low ATP/GTP: 250 μM, high ATP/GTP: 6X-K$_M$ for each dsDNA). The NTase activity of cGAS$^{cat}$ induced by 564 bp dsDNA was normalized by enzyme and substrate concentrations, and used as the reference to calculate the fraction activity for each dsDNA length ($n$ = 3; ±SD). (**E**) The equilibrium-based activation model of cGAS. Diamonds represent basally active cGAS and circles indicate

*Figure 7 continued on next page*

*Figure 7 continued*

active cGAS. Filled shapes and thicker lines indicate dsDNA binding and substrate binding, respectively. Thicker lines and darker shades indicate stronger interactions.

DOI: https://doi.org/10.7554/eLife.39984.020

The following figure supplement is available for figure 7:

**Figure supplement 1.** Context dependent activation of cGAS[FL].

DOI: https://doi.org/10.7554/eLife.39984.021

intracellular NTP levels to increase its dimer population, potentiating its activation. Importantly, increasing the dimeric fraction in the absence of dsDNA would not elicit significant spurious activity, but would instead prime the enzyme for facile activation by reducing the dependence on dsDNA length (*Figure 7E* lower left-hand corner). Another key feature of our equilibrium-based allosteric model is that dsDNA length-dependence is conditional, reconciling conflicting claims regarding the dependence on dsDNA length in activating cGAS (*Andreeva et al., 2017*; *Gao et al., 2013b*; *Kranzusch et al., 2013*; *Li et al., 2013*; *Luecke et al., 2017*; *Zhang et al., 2014*).

## Molecular framework for the dsDNA length-dependent response of cGAS

As the initial receptor in a major inflammatory signaling pathway (*Chen et al., 2016a*), it is critical for cGAS to possess a very stringent noise filtering mechanism. Although cGAS binds dsDNA in a sequence-independent manner (*Gao et al., 2013b*; *Li et al., 2013*; *Zhang et al., 2014*), it uses dsDNA length to distinguish signal from noise (*Andreeva et al., 2017*; *Luecke et al., 2017*). After all, dsDNAs arising from catastrophic conditions are significantly longer than 300 bps (e.g. mitochondrial, genomic, and viral), while short dsDNAs likely indicate minor genome repair and/or resolution of infection (i.e. the viral genome has been degraded). Here, we find that the allosteric coupling mechanism allows cGAS to generate a two-stage noise filter against short dsDNA. For instance, as others have reported (*Andreeva et al., 2017*), we recapitulate here that cGAS binds and dimerizes on dsDNA in a length-dependent manner. Also as reported, we found that dsDNA length-dependent dimerization and binding of cGAS in vitro only gradually changes (*Figures 2–4*; *Andreeva et al., 2017*). However, we found that dsDNA length also grades the enzymatic activity of cGAS (*Figures 3–4*). Thus, combined with the length-dependent complex formation of cGAS dimers (signal recognition), the length-dependent enzymatic activity (signal transduction) would allow cGAS to further differentiate correct pathogenic dsDNA from noise (short dsDNA). Of note, given that dsDNA length-dependence subsides with high concentrations of cGAS, our new model also provides an avenue for how improper clearance of pathogenic or self-dsDNA can induce spurious activity of cGAS leading to auto-inflammatory conditions (*Gao et al., 2015*; *Li and Chen, 2018*).

## The role of cooperativity in initiating and terminating the cGAS pathway

The interactions between cGAS and its ligands (dsDNA and ATP/GTP) display positive cooperativity, a hallmark of allosteric enzymes (*Figures 2–4*). One key feature of a cooperative system is its capacity to amplify and attenuate the output in a switch-like manner (*Monod et al., 1965*; *Sohn and Sauer, 2009*). For instance, when the concentrations of cGAS, dsDNA, and ATP/GTP change by a factor of two, a non-cooperative system would yield a total 8-fold increase in output ($2 \times 2 \times 2 = 8$). However, because cGAS requires dimerization for activity and displays a Hill constant near two in its interaction with both dsDNA and ATP/GTP, the same two-fold change would be further amplified by the exponent of two, leading to a 64-fold amplification in output ($2^2 \times 2^2 \times 2^2 = 64$). Conversely, the same cooperative mechanism would allow cGAS to attenuate its signaling output by the same magnitude with decreasing enzyme and ligand concentrations. Together with the dsDNA-length dependent activity, the cooperativity would enable cGAS to dramatically alter its output according to the changes in input parameters, allowing the initial receptor to dynamically regulate its signaling pathway in a switch-like manner.

## The role of N-domain and human vs. mouse cGAS

Although cGAS$^{cat}$ is sufficient to bind dsDNA and generate cGAMP in vitro, the intact N-domain is crucial for augmenting its function in cells (*Tao et al., 2017*; *Wang et al., 2017*). It has been presumed that the major role of the N-domain is to enhance dsDNA binding (*Lee et al., 2017*; *Tao et al., 2017*). Furthermore, it was proposed that the N-domain promotes the activation of monomeric mouse cGAS by dsDNA (*Lee et al., 2017*). Here, we found that N-domain potentiates the dimerization of cGAS. Our results also indicate that dimerization is necessary for dsDNA-mediated activation by both cGAS$^{cat}$ and cGAS$^{FL}$ (*Figure 5*). It is possible that mouse cGAS operates in a different mechanism than human cGAS. Indeed, it was recently proposed that mouse-cGAS would not depend on dsDNA length as much as human-cGAS for activation, as the former binds short dsDNA more tightly (*Zhou et al., 2018*). However, it was previously shown that both human and mouse-cGAS exhibit similar dsDNA length dependent activation (*Andreeva et al., 2017*). Considering that dsDNA-mediated dimerization is critical for both human and mouse cGAS variants for activation (*Andreeva et al., 2017*; *Li et al., 2013*; *Zhang et al., 2014*; *Zhou et al., 2018*), we propose that our findings are likely general phenomena across different species, and different intrinsic affinity constants caused by diverse primary sequences (*Zhou et al., 2018*) would dictate species-specific experimental observations.

## Comparison with other nucleic acid sensors

Absent-in-melanoma-2 (AIM2) is another major cytoplasmic dsDNA sensor in mammals (*Fernandes-Alnemri et al., 2009*; *Hornung et al., 2009*; *Roberts et al., 2009*). The single most important goal of the AIM2-mediated dsDNA sensing pathway is to induce cell-death, a digital (not tunable) process that does not require a new equilibrium (*Liu et al., 2014*; *Roberts et al., 2009*). Indeed, once assembled on dsDNA, the AIM2 inflammasome does not disassemble and multiple positive feedback loops reinforce the assembly, consequently generating a binary signaling response (*Matyszewski et al., 2018*). By contrast, the cGAS signaling pathway elicits various stress-responses ranging from viral replication restriction to apoptosis, with the signal strength and cellular contexts determining the type of outcome (*Gulen et al., 2017*; *Larkin et al., 2017*; *Li and Chen, 2018*; *Li et al., 2016*; *Liang et al., 2014*; *Tang et al., 2016*; *Yang et al., 2017*). Unlike AIM2, we find here that cGAS can dial its own activity (tunable), providing a molecular framework for eliciting various cGAMP-dependent outcomes. Furthermore, although both AIM2 and cGAS are activated in a dsDNA length-dependent manner, the former assembles into filaments (*Matyszewski et al., 2018*; *Morrone et al., 2015*), while the latter only requires dimerization. Likewise, although cytoplasmic dsRNA sensors preferentially target long duplexes (>500 bps), MDA5 assembles into filaments while RIG-I does not require polymerization for activation (*Linehan et al., 2018*; *Peisley et al., 2011*; *Peisley et al., 2013*; *Ramanathan et al., 2016*; *Sohn and Hur, 2016*). Thus, we propose that the assembly of supra-structures is not universal to host nucleic acid sensors. Rather, it appears that each sensor has evolved unique mechanisms to utilize the length of nucleic acids as a molecular ruler to distinguish self (noise) from nonself (signal).

In closing, our study reconciles the conflicting views on the roles of dsDNA length and the N-domain in activating cGAS. We also provide a mechanistic framework for understanding how cGAS can shape a complex signaling landscape depending on cellular reaction contexts. Future studies will be directed in understanding how this dynamic enzyme operates in conjunction with its downstream and regulatory components to regulate host innate immune responses against cytoplasmic dsDNA.

# Materials and methods

## Reagents

dsDNA substrates and oligonucleotides shorter than 100 bps were purchased from Integrated DNA Technologies (IDT). Longer dsDNAs ($\geq$150 bps) were generated by PCR. The human cGAS cDNA were kindly provided by Dr. Dinshaw Patel. *E. coli* pyrophosphatase was a gift from Dr. James Stivers. The SortaseA (SortA) enzyme was a gift from Dr. Hidde Ploegh. Purity and length of each dsDNA was confirmed by agarose gel electrophoresis. TAMRA- and Cy5-labeled peptides were

purchased from Lifetein. ATP and GTP were purchased from Sigma. GMPcPP and AMPcPP were purchased from Jena Biosciences

## Recombinant cGAS purification

*Protein preparation.* Recombinant cGAS constructs were cloned into the pET28b vector (Novagen) with an N-terminal MBP-tag and a TEV protease cleavage site. Proteins were expressed using 200 µM IPTG at 16°C for overnight in *E. coli* BL21 Rosetta 2. Recombinant cGAS constructs were then purified using amylose affinity chromatography, cation-exchange, and size exclusion chromatography. Tag-free, purified cGAS proteins were then frozen and stored in −80°C with a buffer containing 20 mM Tris HCl at pH 7.5, 300 mM NaCl, 10% glycerol, 5 mM DTT.

*Fluorophore labeling.* The labeling procedure was adapted from (*Guimaraes et al., 2013*). 20 µM MBP-TEV-cGAS-LPETGG-6xHis was incubated with 30 µM SortA, 250 µM fluorophore-peptide in Sortase reaction buffer (50 mM Tris HCl pH 7.5, 150 mM NaCl, 10 mM $CaCl_2$, 5% glycerol, 2 mM DTT) at 25 ± 2°C for 3 hr on a rotator. Reactions were directly applied to Superdex 200 10/300 GL (20 mM Tris HCl pH 8.0, 300 mM NaCl, 2% glycerol, 10 mM BME). Fractions containing cGAS were applied to Ni-NTA. Flow-through was applied to heparin resin and washed with Sizing Buffer. Protein was eluted with Sizing buffer supplemented with 500 mM NaCl. Eluted fractions were adjusted to 20 mM Tris HCl pH 7.5, 300 mM NaCl, 10% glycerol, 5 mM DTT and concentrated.

## Biochemical assays

All experiments were performed at least three times. The fits to data were generated using Kaleidagraph (synergy). Reported values are averages of at least three independent experiments and report errors are standard deviations. All reactions were performed under 25 mM Tris acetate pH 7.4, 125 mM potassium acetate pH 7.4, 2 mM DTT, 5 mM Mg(acetate)$_2$ at pH 7.4, and 5% glycerol at 25 ± 2°C.

*dsDNA binding assays.* Increasing concentrations of cGAS were added to a fixed concentration of fluorescein-amidite-labeled (FAM) dsDNA (5 – 10 nM final). Changes in fluorescence anisotropy were plotted as a function of cGAS concentration and fit to the Hill equation. For competition-based experiments, unlabeled dsDNA was titrated against a fixed population of FAM-dsDNA$_{72}$ and cGAS ([protein] = $K_{D,dsDNA72}$). Changes in fluorescence anisotropy (FA) was plotted against competitor dsDNA concentration and fit to yield IC$_{50}$s.

*FRET-based oligomerization assays.* 60 nM Cy5- and TAMRA-labeled MBP-TEV-cGAS-LPET-GGGQC/K-fluorophore were incubated with TEV protease in cGAS reaction buffer at 25 ± 2°C for 2 hr. Increasing amounts of dsDNAs of different lengths or equimolar concentrations of nucleotides were added to 20 nM cleaved FRET pair, and FRET efficiency was recorded until equilibrium was reached.

*Pyrophosphatase-coupled cGAS activity assay.* cGAS activity was assayed using the pyrophosphatase-coupled assay developed by Stivers and colleagues (*Seamon and Stivers, 2015*) with modifications. Briefly, cGAS was incubated with 50 nM *E. coli* pyrophosphatase, equimolar concentrations of ATP and GTP plus dsDNAs (where indicated) in the reaction buffer. At indicated time points, an aliquot was taken and mixed with an equal volume of quench solution (Reaction buffer minus Mg$^{++}$ plus 25 mM EDTA). Quenched solutions were then mixed with 10 µl malachite green solution and incubated for 45 min at RT. Absorbance at ~620 nm was compared to an internal standard curve of inorganic phosphate to determine the concentration of phosphate in each well. Phosphate concentrations of control reactions devoid of recombinant cGAS were subtracted from reactions containing recombinant cGAS. Apparent catalytic rates were calculated from the slopes of control-subtracted phosphate concentrations over time. Reported rates were halved to reflect pyrophosphate production. Average values are listed in Tables.

*nsEM.* Experiments were conducted using a Philips BioTwin CM120 (FEI) as described previously (*Morrone et al., 2015*).

## SAXS data collection and analysis

SAXS data was collected on the BIOSAXS 2000 (Rigaku) at the X-ray facility of the Department of Biophysics and Biophysical Chemistry at Johns Hopkins School of Medicine. Data was collected on at least three different concentrations for each sample. SamplesBi with scatter showing significant

inter-particle effects were omitted from data analysis. Buffer-subtracted scatter was processed in Scatter (*Mylonas and Svergun, 2007*; *Petoukhov et al., 2012*; *Petoukhov and Svergun, 2013*) and with the ATSAS package (*Mylonas and Svergun, 2007*; *Petoukhov et al., 2012*; *Petoukhov and Svergun, 2013*). Particle dimensions were compared between guinier analysis and real-space fitting of the scatter to ensure internal consistency of the data and fits. Estimates of average and relative molecular weights of each sample were estimated using porod volumes (*Mylonas and Svergun, 2007*; *Petoukhov et al., 2012*; *Petoukhov and Svergun, 2013*) and mass-normalized $I_0$ values. The distribution of monomeric and dimeric species was calculated using SAXS-estimated molecular weights and OLIGOMER. IN OLIGOMER, crystal structures of monomeric cGAS and dimeric cGAS were used as a reference (PDB ID: 4LEV).

## Acknowledgement

We thank the Sohn lab members, Drs James Stivers, Mario Amzel, Scott Bailey, and Herschel Wade for helpful discussion. We thank Mariusz Matyszewski for the assistance in nsEM experiments. This work was supported by American Cancer Society Research Scholars Grant (DMC-RG-15 – 224), National Institutes of Health (R01GM129342A1), and Johns Hopkins School of Medicine Synergy Award to JS.

## Additional information

### Funding

| Funder | Grant reference number | Author |
|---|---|---|
| National Institutes of Health | R01GM129342A1 | Jungsan Sohn |
| American Cancer Society | DMC-RG-15-224 | Jungsan Sohn |

The funders had no role in study design, data collection and interpretation, or the decision to submit the work for publication.

### Author contributions

Richard M Hooy, Conceptualization, Data curation, Formal analysis, Investigation, Methodology, Writing—review and editing; Jungsan Sohn, Conceptualization, Formal analysis, Funding acquisition, Writing—original draft, Project administration, Writing—review and editing

### Author ORCIDs

Richard M Hooy http://orcid.org/0000-0001-8917-0249
Jungsan Sohn http://orcid.org/0000-0002-9570-2544

### Decision letter and Author response

Decision letter https://doi.org/10.7554/eLife.39984.024
Author response https://doi.org/10.7554/eLife.39984.025

## Additional files

### Supplementary files

• Transparent reporting form
DOI: https://doi.org/10.7554/eLife.39984.022

All data generated or analysed during this study are included in the manuscript and supporting files.

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
