## [Decision Letter]

Thank you for submitting your article "The allosteric activation of cGAS underpins its dynamic signaling landscape" for consideration by *eLife*. Your article has been reviewed by Philip Cole as the Senior Editor, a Reviewing Editor, and two reviewers. The following individuals involved in review of your submission have agreed to reveal their identity: Sarah E Walker (Reviewer #2); Claus D. Kuhn (Reviewer #3).

The reviewers have discussed the reviews with one another and the Reviewing Editor has drafted this decision to help you prepare a revised submission.

Summary:

In the manuscript, "The allosteric activation of cGAS underpins its dynamic signaling landscape", Hooy and Sohn clarify dimerization properties that contribute to the function of cyclic-G/AMP synthase (cGAS). The results indicate that cGAS can dimerize independently of substrate binding with low affinity, but that the affinity for dimerization is increased by binding to dsDNA or to a lesser degree by the cG/ATP substrates or nonhydrolyzable analogues. Full-length cGAS had an even higher affinity for dimerization. Importantly, the authors use a nice PP_i_ exchange assay to follow cGAMP activity without use of labeled substrates that would have otherwise precluded monitoring steady-state kinetics. These experiments suggest that the length of dsDNA not only affects the apparent affinity (K_act_) for dsDNA, but also the maximal rate of the reaction. Analogous experiments for ATP/GTP substrate turnover indicate that longer DNA is also required for maximal activity and apparent affinity for substrates, with a maximal change in catalytic efficiency of ~8-fold resulting from increasing DNA length. A dimerization defective mutant revealed that dimerization, and not dsDNA length is responsible for increased *k*_cat_. Negative staining EM seems to support the idea that dimers form stochastically on the dsDNA rather than forming a ladder structure.

This was a well-written paper and a great example of rigorous, quantitative biochemistry revealing complex molecular mechanisms (with a few exceptions, see later). The authors' hypothesis that dsDNA length increases the fraction of active cGAS dimers without imposing a structural arrangement, is highly interesting. Especially with Figure 7, they provide a convincing, novel explanation for how all parameters influencing cGAS activation can be linked.

However, their experiments do not sufficiently substantiate these novel claims (which is also a prominent part of their abstract). Some key issues need to be addressed to firmly establish the proposed model versus that of the Andreeva et al. (2017) model.

Essential revisions:

1) The EM images need to be substantially improved – they are currently not convincing enough. A more quantitative approach to EM image analysis is necessary. For example, the authors say "The clusters formed by K394E variants on dsDNA appeared smaller than those formed by wild-type when the protein was in excess over dsDNA." Stating 'appeared smaller' isn't very quantitative. Is it possible to measure the clusters or clarify the size disparity by use of a scale bar on each panel being compared?

2) The connection between the mechanistic study of the authors and the differential activation of cGAS in vivo is not apparent until at the end. We suggest that the novel model of Figure 7, be placed in context of the in vivo signaling situation throughout the manuscript. This connection should be made as a clear thread running throughout the manuscript, to make the findings accessible to the broad and diverse readership of *eLife*. In addition, the authors are advised to place the work of Zhou et al. (2018) in context with the findings of their study.

3) Figure 2 and Figure 3C, 3E: The claim of a dsDNA activation jump with dsDNA >300 bps cannot be made with the existing data points. The experiments lack dsDNAs between 172 bps and 339 bps. For claiming a step-like activation or 300 bps "jump" one would have to include more dsDNAs with lengths between 172 and 339.

4) Figure 3E: The authors state that they were unable to saturate the reaction with AMP/GTP concentration > 2mM but show data only up to 1 mM. They should show data with higher substrate concentrations or amend the statement.

---

## [Author Response]

Essential revisions:1) The EM images need to be substantially improved – they are currently not convincing enough. A more quantitative approach to EM image analysis is necessary. For example, the authors say "The clusters formed by K394E variants on dsDNA appeared smaller than those formed by wild-type when the protein was in excess over dsDNA." Stating 'appeared smaller' isn't very quantitative. Is it possible to measure the clusters or clarify the size disparity by use of a scale bar on each panel being compared?

We added scale bars in each panel. We also measured representative particles and indicated their sizes. Figure 6 is thus expanded to include zoomed-in images (Figure 6—figure supplement 1). Additionally, we discuss how the estimated particle sizes correspond to those of monomeric and dimeric cGAS^cat^ and cGAS^FL^ in subsection “cGAS dimers arrange randomly on dsDNA”.

2) The connection between the mechanistic study of the authors and the differential activation of cGAS in vivo is not apparent until at the end. We suggest that the novel model of Figure 7, be placed in context of the in vivo signaling situation throughout the manuscript. This connection should be made as a clear thread running throughout the manuscript, to make the findings accessible to the broad and diverse readership of eLife. In addition, the authors are advised to place the work of Zhou et al. (2018) in context with the findings of their study.

We added figures describing the allosteric framework for the activation of cGAS in Figure 2A and 2B (we thus rearranged Figure 2). Please see the subsection “cGAS behaves like an allosteric enzyme”. We then frequently refer to these figures thereafter.

We also discuss the findings from Zhou et al. in light of our studies in subsection “The role N-domain and human vs. mouse cGAS”.

3) Figure 2 and Figure 3C, 3E: The claim of a dsDNA activation jump with dsDNA >300 bps cannot be made with the existing data points. The experiments lack dsDNAs between 172 bps and 339 bps. For claiming a step-like activation or 300 bps "jump" one would have to include more dsDNAs with lengths between 172 and 339.

We added data showing 177 and 249 bp dsDNA in these figures. Importantly, our results demonstrate that both apparent binding affinity of dsDNA (K_act_ and K_FRET_) and the extent activation (*k*_max_) change according to dsDNA length. Our results in turn demonstrate that the overall signaling efficiency (i.e. signal recognition and transduction) changes more dramatically over different dsDNA lengths than either parameter alone. To underscore these observations we added: “For instance, the overall signaling efficiency changes by nearly 100-fold between 24 to 339 bp dsDNA, while either binding or maximal activity alone change by 10-fold (Figure 3D, see also Figure 3—figure supplement 2A-D).” in subsection “dsDNA length regulates the extent of activation”.

4) Figure 3E: The authors state that they were unable to saturate the reaction with AMP/GTP concentration > 2mM but show data only up to 1 mM. They should show data with higher substrate concentrations or amend the statement.

We added data showing up to 2 mM ATP/GTP for dsDNA-free cGAS^cat^ in Figure 3—figure supplement 1B. cGAS^cat^ does appear to saturate with 2 mM substrates. However, given the large error with high [ATP/GTP] for cGAS^cat^, we report *k*_cat_/K_M_ values for these experiments. On the other hand, cGAS^FL^ without dsDNA did not show any saturation (Figure 4—figure supplement 3E). We also revised the corresponding section in subsection “dsDNA length regulates formation of the enzyme-substrate complex (KM) and the turnover efficiency (*k*cat) of cGAS”: “Without dsDNA, cGAS^cat^ showed measurable NTase activities (Figure 3—figure supplement 3B).”